

# Radial growth of *Picea abies* is controlled by joint effects of temperature and nutrient availability at the lower part of treeline ecotone

Hana Kuželová[1], Tomáš Chuman[1], Jelena Lange[1], Jan Tumajer[1], Václav Treml[1]

[1]Department of Physical Geography and Geoecology, Faculty of Science, Charles University, Albertov 6, 12800 Prague, Czech Republic

*Correspondence to*: Václav Treml (treml@natur.cuni.cz)

**Abstract.** Treeline ecotones in complex mountain landscapes are exposed to pronounced differences in irradiation and soil nutrient availability. Different amounts of nutrients and direct solar energy can influence tree stem growth resulting in variation of growth rates and growth phenology across lower parts of treeline ecotone. We hypothesized that at two contrasting sites located on north and south-facing slopes, differences in nutrient availability outperform temperature differences in modulating stem growth rates while growth phenology is driven by the course of seasonal temperature. To test this hypothesis, we compared the growth phenology and kinetics of *Picea abies* in the lower part of treeline ecotone between a north-facing slope with relatively nutrient-rich soils and a south-facing slope with nutrient-poor soils. We analysed intra-annual wood formation and its response to ambient climate, as well as soil and air microclimate and soil and needle nutrient content. Our results showed that thermal differences in treeline ecotones between south and north-facing slopes in temperate mountains are small but nontrivial involving higher daytime temperatures at south-facing slopes and longer irradiation of north-facing slopes during the middle part of growing season. The timing of growth onset and maximum growth rate were almost identical on both slopes. Accordingly, annual stem growth at both sites was most sensitive to the meteorological conditions at the start of the growing season and around the summer solstice. However, the absolute growth rate was higher on the north-facing slope, consistent with a higher availability and content of base cations in the soil and the needles. Our results suggest that temperature governs growth phenology at the lower part of the treeline ecotone, but nutrient availability modulates the growth rate in the peak season when temperature no longer limits cambial activity. We conclude that the effect of nutrient availability can be superior to the effect of slope aspect for stem growth rates of *Picea abies* located in the lower part of treeline ecotone in temperate mountain range.

## 1 Introduction

In cold environments, tree stem growth is tightly linked to temperature oscillations during the growing season (Rossi et al., 2016; Körner, 2021), which is reflected in the seasonal phenology of wood formation, that is, cambial division and xylem differentiation (Cuny et al., 2015). There are minimum temperature thresholds for both photosynthesis and the processes



linked to the investment of non-structural carbohydrates into developing xylem, the latter being lower and thus representing
the ultimate limitation of stem growth (Fatichi et al., 2019). This is apparent along elevation transects where both non-
structural carbohydrates and nutrient contents in twigs, leaves and sapwood increase towards the treeline as a result of
increasing sink limitation of the growth (Hoch and Körner, 2012; Fajardo and Piper 2017; Doležal et al., 2019). While no
exceptions from pure sink limitation of tree growth have been found for broadleave treelines in the southern hemisphere
(Fajardo and Piper 2017), some studies examining conifers in permafrost zone have highlighted the important role of nutrient
availability in co-limiting stem growth (Sullivan et al., 2015; Dial et al., 2022). Although this is probably not generally valid
for conifer treelines outside the permafrost zone (Hagedorn et al. 2020; Körner, 2021), the nutrient and moisture co-
limitation of tree growth can play a role at local upper tree limits (boundaries of realized niche) or in the lower part of
treeline ecotone, several tens to hundred meters below treeline (Möhl et al., 2018, Körner and Hoch, 2023).
The low-temperature limit of tree growth at its cold range boundary is evidenced by growth resumption after exceeding a
certain temperature threshold, as shown both by warming/cooling experiments (Gričar et al., 2006; Lenz et al.; 2013) and by
observations in natural treeline settings (Körner and Hoch, 2006; Rossi et al., 2007). Indirectly, the prevailing low-
temperature limitation of tree growth at cold sites is supported by similar thermal limits of global treelines, i.e., cold margins
of the tree life form distribution (Körner and Paulsen, 2004). Furthermore, tree-ring chronologies from treelines are
significantly correlated with growing season temperature (Chagnon et al., 2023), and calibrated thermal limits of wood
formation models agree with those based on direct or experimental observations (Tumajer et al., 2021).
Recently, tree growth in cold biomes tends to accelerate in some areas, which has often been attributed to warming and an
extension of the growing season (Shi et al., 2020). However, in forest stands near treeline, observed growth enhancement has
been connected to increased nitrogen supply in some regions (Kolář et al., 2015; Möhl et al., 2018; Etzold et al., 2020).
Sullivan et al. (2015) showed better performance in shoot, stem and root growth at microsites relatively richer in nitrogen at
the Arctic treeline due to warmer soils and a higher snowpack accelerating nutrient cycles (Dawes et al., 2017). Not only
nitrogen (N) but also phosphorus (P, and namely the stoichiometry of N and P) was suggested as a limiting factor of tree
occurrence at some treeline ecotone sites in the Himalayas (Müller et al., 2017). Apart from N and P, the role of other
nutrients, especially base cations, has been largely neglected at tree stands near their cold distribution margins. Recently,
there is a growing body of literature showing that base cations can play a vital role under certain conditions in limiting tree
growth in the treeline ecotone (Drollinger et al., 2017) or in montane forests (Körner, 2022; Oulehle et al., 2023).
Local evidences of growth enhancements at sites relatively enriched by N and P are not necessarily in conflict with the
ultimate role of low temperature limitation of tree growth at treeline (Körner, 2012) and with observations of increasing
nutrient concentrations in leaves with elevation near cold margins of tree distribution (Fajardo and Piper, 2017). They rather
suggest that nutrient availability together with low temperature might co-determine growth dynamics at upper tree limits
whose position lags behind the pace of warming (Fatichi et al., 2019). As many current upper tree limits cannot follow the
upward shift of isotherms (Körner and Hiltbrunner, 2024), the understanding the importance of climate, nutrient availability,



and their interaction as determinants of the growth dynamics becomes critical in a period of unprecedented impacts of
warming on mountain ecosystems.
In a complex mountain relief, a high variation in topoclimate and soil conditions can create a heterogenous mosaic of sites
differing in local surface temperature (Körner, 2012; Jochner et al., 2017; Kuželová et al., 2021) as well as nutrient content
(Liptzin et al., 2013, Mayor et al., 2017). Probably the best-known topoclimatic effect is the so-called slope exposure
phenomenon, which suggests that south-facing slopes in the northern hemisphere outside the tropics are warmer than north-
facing slopes, and vice versa in the southern hemisphere (Körner, 2012). This phenomenon is less pronounced on forested
slopes (Paulsen et al., 2001). Sites on opposite slopes might differ not only in insolation and surface temperature but also in
nutrient availability, as surface temperature might influence litter decomposition through surface moisture and snow melt
patterns (Dawes et al., 2017; Ellison et al., 2019; Stark et al., 2023).
Therefore, an experimental design employing opposite south-facing (S-slope) and north-facing slopes (N-slope) at the same
elevation provides an excellent natural settings where organisms are exposed to pronounced differences in incoming solar
radiation under a similar macroclimate. Such a design has previously been used to test various ecological hypotheses related
to tree growth and its limiting factors (Rossi et al., 2007; Moser et al., 2010; Tyagi et al., 2023). In this study, we use an
opposite slope aspect design in a complex research setting, which allows us to cover multiple site properties that potentially
influence tree growth performance in the lower part of the treeline ecotone. We carried out detailed observations of Picea
abies trees in terms of intra-annual wood formation and inter-annual climate-growth responses of radial stem growth together
with analyses of site thermal and moisture properties and soil and foliar nutrient content. We agree with and test the general
assumption that at the lower part of treeline ecotone, the crucial phases of tree growth, such as growth resumption and timing
of the peak growth rate, are driven by thermal and solar constraints (Rossi et al., 2006b; Rossi et al., 2007). However, we
hypothesize that the absolute growth rate in the middle part of the growing season is influenced by nutrient availability,
given the positive effect of nutrient availability on absolute tree growth that has been shown at some cold-limited sites (Möhl
et al., 2018; Sullivan et al., 2015).

## 2 Material and Methods

### 2.1 Study area

Our study focuses on Picea abies [L.] Karst. growing on N and S-slopes in the treeline ecotone in the Krkonoše Mts, Czech
Republic (50°43′N, 15°40′E, site elevation 1250 m a.s.l., Figure 1). The Krkonoše Mts, with the highest peak Mt Sněžka
(1602 m a.s.l.), are characterized by high annual precipitation sums (on average 1400–1600 mm) and a mean annual
temperature of 0.5°C in the uppermost locations (Metelka et al., 2007). Snow cover in the treeline ecotone lasts from
November until May, with a maximum snow depth of about 2 m (Metelka et al., 2007). The treeline ecotone is situated at
elevations ranging from 1250 to 1450 m.





Two sites on opposite slopes (N-slope, S-slope) were established in the lower part of the treeline ecotone in the Bílé Labe
valley (Figure 1). The canopy cover at both sites was 20 %, and the tree height of adult individuals ranged between 8 and 13
m. Both sites are located on steep slopes (inclination between 20° and 30°) with frequent patches of screes. The N-slope site
is located on the transition between gneiss and mica-schist (upper part) and granodiorites (lower part). The S-slope site is
underlaid by granites. Skeletic Leptosols and Skeletic Podzols are the dominant soil types on the S-slope, and the same soil
types prevail on the N-slope, with patches of Histic Skeletic Podzols.

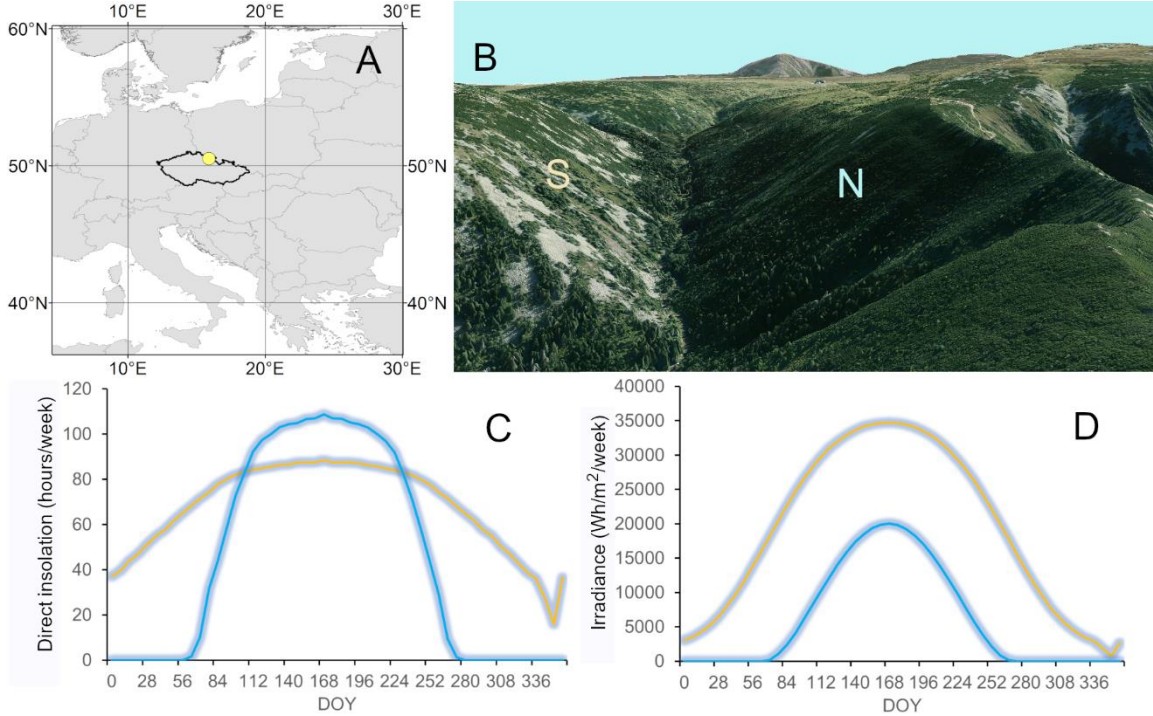


**Figure 1: (A) Location of the study area in Europe; (B) View from the west on the study sites (modelled by ArcScene; ESRI, 2020).**
**Modelled duration of direct insolation (C) and irradiance (D) plotted against the day of year (DOY) on the north-facing (N, blue)**
**and south-facing (S, orange) sites.**
**2.2 Microclimatic monitoring**
We measured air and soil temperature and soil water potential to characterize site micro-climatic conditions from 2012 to
2015 (to 2014 for soil water potential). Air temperature was recorded using sensors in radiation shields hanging in the tree
crown approximately 7 m above the ground (one sensor at each site). Three sensors recorded the soil temperature and soil
water potential of the root zone (mineral soil, −10 cm depth) per site at microsites fully shaded by tree crowns. We used
gypsum block soil water potential sensors to measure available soil moisture (measuring range 0 to -2 MPa, Delmhorst, EMS
Brno). Air and soil temperatures and soil water potential were measured and stored at 1-hour intervals. Both air and soil
temperature sensors have an accuracy of ± 0.2°C (www.emsbrno.cz).





For each year and site, we calculated various variables that characterize the intra-annual meteorological patterns that are
important for woody plants. Following Körner and Hiltbrunner (2018), we used soil temperature to define the duration of the
meteorological growing season, i.e., the period when meteorological conditions potentially permit wood formation. We used
the continuous period where soil temperature values were above 3.2°C, roughly corresponding to a mean weekly air
temperature of 0°C (Körner and Paulsen, 2004). For both sites and for all years, we identified dates of the start, the end, and
the duration of the meteorological growing season. Additionally, we computed mean air and soil temperature for the period
June to September per site and year. Lastly, we calculated degree days by integrating mean daily air temperatures exceeding
5°C for each site and year.
Site insolation was estimated using the ArcGIS Solar radiation tool (ESRI, 2020). Based on a digital elevation model with 5
m resolution, we modelled the duration of direct insolation in hours and the solar irradiance (W/m-2) for both sites and each
day of the year. For each site, the insolation depended on the sun's position, slope aspect and inclination, and the location of
surrounding ridges and peaks that shaded the sites for certain parts of the day and year.

**2.3 Soil and foliar nutrient analyses**

Soil samples were collected in two campaigns in October 2013 and October 2023. In the first campaign, we took two soil
samples, and in the second campaign, we took three soil samples from the topmost 10 cm of the mineral soil at each site.
Each sample from each campaign was pooled from five subsamples distributed randomly over each study site and properly
mixed. Air-dried soils were sieved to remove the size fraction > 2 mm. Samples were analysed for exchangeable pH
(CaCl2), cation exchange capacity (CEC), total soil N, soil organic carbon (Cox), and plant-available concentrations of Ca,
Mg, K and P with Mehlich III extraction solution (Mehlich, 1984). Soil samples were analysed in the accredited laboratory
of the Research Institute for Soil and Water Conservation, Prague. Differences between sites in measured soil variables were
tested using the Kruskal-Wallis test implemented in R (R Development Core Team, 2023). For this purpose, we merged
samples from both campaigns.
To compare the nutrient concentrations in needles between sites and to validate whether they reflect nutrient availability in
soils, we collected current-year and previous-year needles from six trees at each site in October 2023. The long lag between
the sampling of wood formation (2012-2014) and the sampling of foliar macroelements (2023) might influence absolute
values of the determined elements because of their interannual variability, but not the difference between sites, which
remains constant (Novotný et al., 2018). Branches from the upper part of the crown were cut using a telescopic long-reach
pruner. Current- and previous-year needles were sampled from all cut branches and pooled per site. Pooled samples of 1000
current-year and 1000 previous-year needles were dried and then analysed for the content of main macro- and
microelements. The analyses of the main macro- and microelements followed a standard ICP Forests protocol (Rautio et al.,
2020). The foliage K, Ca, Mg and P was determined using ICP-OES after needle decomposition in a microwave oven. The
total S and N content was analysed using the Leco CNS element analyser (Elementar Analysensysteme GmbH, Germany).





## 2.4 Wood formation

Six (2012) to eight trees (2013-2014) at each site were monitored in terms of wood formation ("xylogenesis") over the three growing seasons (Table S1), which overlapped with the period of microclimatic measurements. A new set of healthy and dominant/co-dominant trees was selected for sampling each season to avoid possible impacts of previous year's sampling on ongoing cambial activity, e.g., by the formation of traumatic resin ducts. Wood microcores were sampled using a Trephor puncher (Rossi et al., 2006a) at a stem height of $1 \pm 0.2$ m. Each sample contained the xylem of the current year, the cambial zone, the phloem, and one or more previous complete annual rings. The distance between adjacent sampling points on a stem was always greater than 3 cm to avoid effects of sampling on wood formation. Sampling intervals ranged from 7 to 10 days during the period from April to October, which significantly exceeds the typical duration of a growing season in a treeline environment (Treml et al., 2015). Once sampled, the microcores were immersed in a formaldehyde-ethanol-acetic acid fixative. The laboratory procedures followed Gričar et al. (2006). The microcores were dehydrated using a successive series of ethanol and xylol-substitute and were then embedded in paraffin. 12-μm-thick cross sections were cut using a rotary microtome. The paraffin was removed, and samples were dehydrated using a successive series of xylol-substitute and ethanol solutions with descending/ascending ethanol concentrations. The cross sections were then stained with safranin and astra blue and mounted on permanent slides using Canada balsam.

The cells in the following wood phenological phases were counted for each cross section under 400–500x magnification using an optical microscope following Rossi et al. (2003): cells in the cambial zone, enlarging cells, wall-thickening cells, and mature cells. The number of cells in each developmental stage was counted in three radial files and subsequently averaged. The number of cells in the preceding tree ring was counted for three radial files and averaged. For each tree, the start and end date of each developmental phase (onset of cambial activity, enlarging phase, cell wall thickening phase, mature phase), and the overall duration of cambial activity were determined according to Rossi et al. (2007).

The counts of cells developed over the course of the growing season were fitted by a Gompertz function using the R package CAVIAR (Rathgeber et al., 2018). Next, the following parameters were determined from the Gompertz equation for each tree: the maximum daily cell production rate, the day of maximum cell production rate (both called critical dates, Rathgeber et al., 2018), and the mean daily production rate in the period when 90 % of cells were formed. Between-site differences of critical dates and production rates were tested using the Kruskal-Wallis test implemented in R (R Development Core Team, 2023).

Logistic regressions were calculated to identify temperature thresholds at which the wood formation resumes (Rossi et al., 2008), with active/inactive wood formation as the explained binary variable and the 7-day backward mean soil and air temperature as explanatory variables. Only observations before the summer solstice were considered (Treml et al., 2019). The start of the active wood formation was alternatively defined by the occurrence of the first new cells in the cambial zone or the first enlarging cells. All calculations were performed in R (R Development Core Team, 2023).



## 2.5 Climate-growth relationships of tree-ring chronologies

Wooden cores were extracted at 1 m stem height from 30 randomly selected dominant and co-dominant individuals of Picea abies at each site in October 2013 using an increment borer (5 mm in diameter). Following standard laboratory procedures (fixation of cores to wooden supports, air-drying, sanding), tree-ring widths were measured using the WinDendro system (scanner and software with semi-automatic ring detection) (Regent Instruments, 2021). The resulting tree-ring series were visually and statistically cross-dated using PAST 5 software (Knibbe, 2013; Speer, 2010). We focused on high-frequency (interannual) growth variability preserved in tree-ring data. Therefore, tree-ring series were standardized with a cubic smoothing spline with a 40-year window length at a 50% frequency cutoff, and the autocorrelation was removed using autoregressive modelling (Cook and Peters, 1981). The tree-ring chronology for each site was built by averaging tree-ring series of individual trees. We calculated Pearson correlations between tree-ring chronologies and climatic time series with daily resolution and run the correlation analysis for time spans from ten to thirty consecutive days (Jevšenak, 2019). Daily climatic data (mean daily temperature, daily precipitation totals) from the nearest meteorological station Labská/Vrbatova bouda (1320 m a.s.l., 8 km westwards from our sites) were available and used from 1961 after filling data gaps using the neighboring station. Before calculating correlations, temperature data were standardized in the same way as tree-ring data (Ols et al., 2023). Tree-ring and temperature data sstandardization and climate-growth correlations were performed using the dplR (Bunn et al., 2023) and dendroTools (Jevšenak and Levanič, 2018) R packages (R Development Core Team, 2023).

## 3 Results

### 3.1 Solar radiation and temperature differences

Irradiance was considerably higher on the S-slope than on the N-slope over the entire year (Fig. 1D). The N- slope was characterized by more than five months in winter without direct insolation (Fig. 1C). The duration of direct insolation was accordingly longer on the S-slope except for the period between end of April and mid August when the weekly duration of insolation was longer on the N-slope, reflecting the effect of the complex topographical setting (Fig. 1B, 2A, 2B). Mean air temperature during the main growing season (June-September) was about 0.1 K higher at the S-slope in most years, but this difference was smaller than the measurement error (Table 1). Similarly, degree days were slightly higher at the S-slope (Table 1). Interestingly, differences in air temperature during the main part of the growing season showed a pronounced daily pattern with a warmer S-slope during the day and a warmer N-slope at night (Fig. 2D).

At both sites, soil temperature oscillated under the snowpack close to 0°C usually until the day of year (DOY) 110-120 (ca. end of April) and then abruptly increased (Fig. 2A). Soils tended to be cooler on the S-slope during the winter, possibly due to deeper freezing under a thinner snowpack. Soil temperature was higher on the S-slope at the very beginning and towards the end of the growing season, while soils on the N-slope tended to be warmer in the peak growing season (Fig. 2C), which roughly corresponds to the period when daily direct insolation is longer on the N-slope (Fig. 1C) . As a result, mean soil



temperature was slightly warmer (0.2-0.3 K) at the N-slope over the June-September period in most years, with differences

again being close to the measurement error, except in 2015 when the S-slope was substantially warmer (Table 1). There was

no systematic pattern in the duration of the meteorologically-defined growing season length (Table 1).

Both sites exhibited several periods with significant negative soil water potentials (Fig. S1), occurring mostly in the summers

of 2013 (both N- and S-slope) and 2014 (N-slope) but also in the winters of 2012 (N-slope) and 2014 (S-slope).

**Figure 2: (A) Daily means of soil (bold lines, bottom) and air temperature (thin lines, top) for the north- (blue) and south- (orange) facing slope for the period 2012-2015. (B) Differences between south- and north-facing slopes (temperature south minus temperature north) smoothed by a 5-day moving average 2012 – 2015 for mean daily air temperature and (C) the same for mean daily soil temperature. (D) Differences in the course of daily air temperature (temperature south – temperature north; hourly interval) for June-August 2012-2015.**





**Table 1: Thermal characteristics of the growing season on the north (N-slope) and south-facing (S-slope) site calculated based on**
**on-site measurements.**

| Year | Site | Growing season duration (days) | Mean Air T in June – September (°C) | Mean Soil T in June – September (°C) | Degree days exceeding 5 °C |
|---|---|---|---|---|---|
| 2012 | N-slope | 145 | 11.26 | 9.96 | 1023 |
| | S-slope | 167 | 11.34 | 9.81 | 1054 |
| 2013 | N-slope | 192 | 10.88 | 9.57 | 906 |
| | S-slope | 192 | 10.99 | 9.32 | 929 |
| 2014 | N-slope | 189 | 11.14 | 9.78 | 978 |
| | S-slope | 158 | 11.05 | 9.48 | 987 |
| 2015 | N-slope | 195 | 12.09 | 8.81 | 1036 |
| | S-slope | 196 | 12.18 | 9.59 | 1062 |
| Mean (±SD) | N-slope | 180±20 | 11.3±0.4 | 9.5±0.4 | 985±50 |
| | S-slope | 178±16 | 11.4±0.5 | 9.6±0.2 | 1008±54 |

**3.2 Nutrient content in soil and foliage**
Soils at both sites were strongly acid with low CEC, P content and plant-available base cations except for K (Fig. 3, Table
S2). The concentrations of base cations were systematically higher on the N-slope than on the S-slope. Statistically
significant differences were detected for CEC and Mg (Fig. 3). The concentration of Ca was below the detection limit for
half of the samples. The content of Cox and N was high, with a favourable C/N ratio at both sites.
In line with soil nutrient analysis, foliar macroelements were higher on the N-slope than on the S-slope both in current- and
previous-year needles (Table 2). For the current-year needles, the concentrations of base cations (Ca, K, Mg) were about 25-
29% higher on the N-slope. The content of P and N was also substantially (16-21%) higher on the N-slope (Table 2).

**Table 2: Foliar nutrients in mg/kg of dry matter. Samples were pooled from six trees at each site.**

| Sample | Ca | K | Mg | P | $N_{total}$ | $S_{total}$ |
|---|---|---|---|---|---|---|
| **N-slope current year** | 3944 | 7059 | 1177 | 1456 | 1.54 | 1040 |
| **S-slope current year** | 2785 | 5248 | 849 | 1213 | 1.32 | 833 |
| **N-slope previous year** | 4954 | 5913 | 1064 | 1245 | 1.64 | 1125 |
| **S-slope previous year** | 4608 | 4650 | 898 | 1039 | 1.31 | 887 |






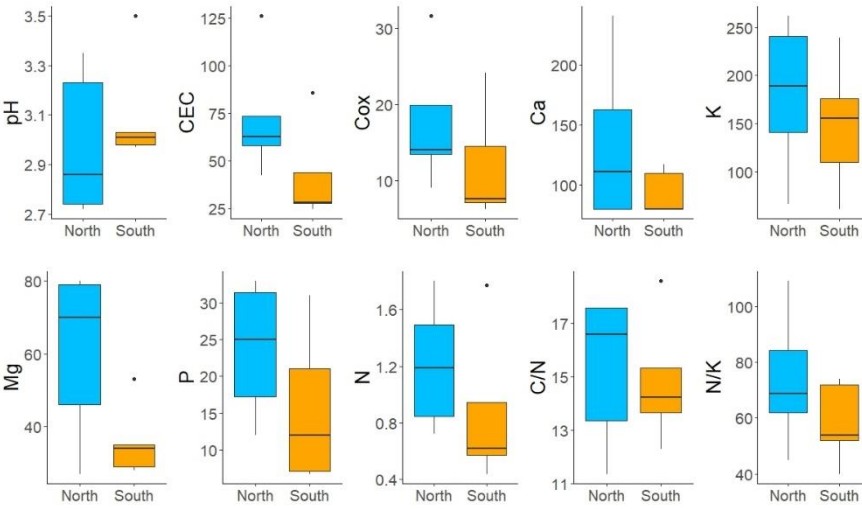


Figure 3: Differences in soil characteristics between south- (orange) and north-facing (blue) slopes. Variables: exchangeable pH, cation exchange capacity (CEC) (mol+/100g), soil organic carbon (Cox) (%), plant-available concentrations of Ca, Mg, K, P with Mehlich III extraction solution (mg/kg), total soil N (%), and ratios of C/N and N/K. The analytical results for Ca that were below the detection limit were replaced by 4/5 of the detection limit.

### 3.3 Wood formation

The N-slope exhibited a higher number of newly formed cells each year, with the greatest difference compared to the S-slope found in 2014 (Fig. 4, Fig. S2). Although the difference was systematic across years and all parts of the growing season, it was statistically non-significant due to the limited number of sampled trees and natural between-tree variability in xylogenesis. The higher number of cells on the N-slope resulted from consistently higher mean and maximum cell formation rates, i.e., faster stem growth; the difference was statistically significant in 2014 ($p < 0.05$) (Fig. 4, Fig. S3). Note that while the tree age distribution was comparable between sites in 2012 and 2014, trees on the N-slope were about 50 years older in 2013 (Table S1). The higher cell formation rates at the N-slope were consistent with significantly higher mean basal area increments at the N-slope compared to S-slope considering the entire lifespan of trees (Fig. S4).

Critical dates of wood formation, such as dates of the beginning, peak and end of wood formation did not show any consistent pattern, and no difference was statistically significant (Fig. S3). The variability in critical dates among trees on the S-slope was usually higher than on the N-slope. Similar to critical dates derived from the Gompertz equation, there were no consistent differences in dates of cell phenological phases between sites based on raw cell development data (Fig. S5). It is worth mentioning that the duration of cell wall thickening was significantly longer on the N-slope in 2013 and 2014.

Logistic regressions with growth resumption indicated by the first enlarging cells (binary response variable) and soil temperature (predictor) better fitted the data than regressions with cambial division (active/inactive - binary response variable) and air temperature (predictor) (Fig. S6). Growth onset represented by the occurrence of the first cambial cells occurred at 3.3 and 3.9 °C soil temperature and 3.6°C and 4.5°C air temperature on the S-slope and N-slope, respectively. In



contrast, thresholds for the first enlarging cells were very similar at both sites and occurred at 4.7 °C of soil temperature at
both sites and 6.4°C and 6.3°C of air temperature on the S-slope and N-slope, respectively.

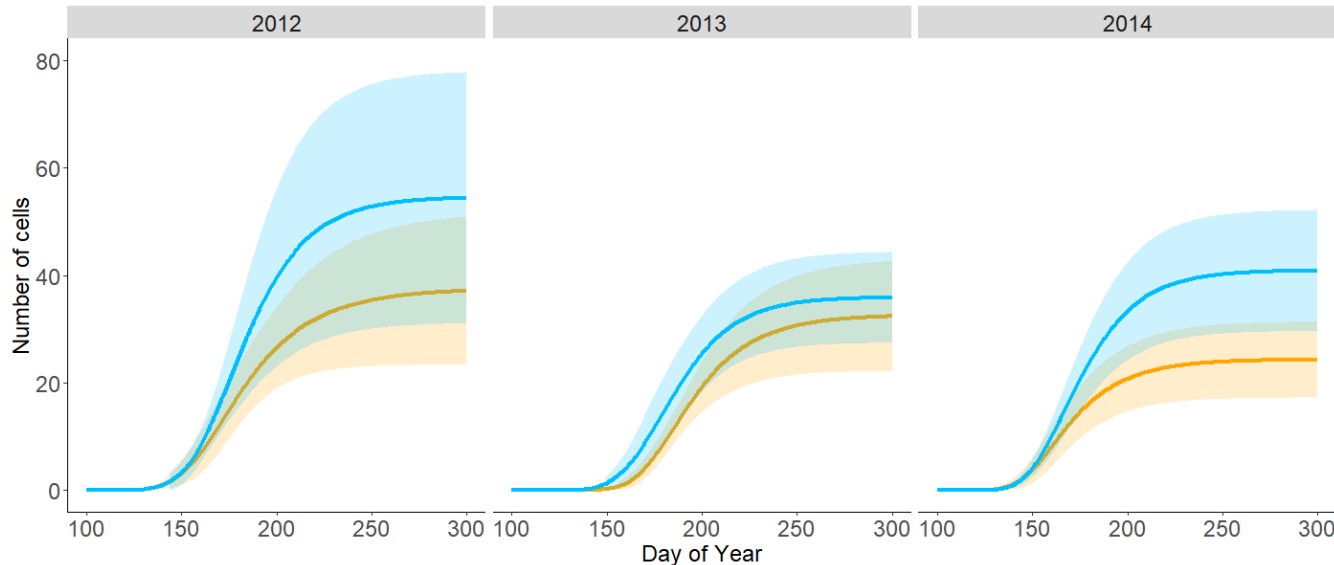

**Figure 4: Number of tracheids in the newly developing annual tree ring for 2012-2014 for the north- (blue) and the south-facing**
**(orange) site. Graphs show data fitted by the Gompertz function. Buffers denote 95 % confidence intervals.**

**3.4 Climate response of tree-ring width chronologies**
Both sites showed two prominent periods with significant temperature-growth correlations, potentially indicating crucial
parts of the season for annual ring width formation – the beginning of the growing season centered around DOY 125 (first
week of May) and the peak growing season centered around DOY 180 (end of June, Fig. 5). Positive temperature-growth
correlations were slightly stronger on the N-slope, with the maximum correlation coefficient exceeding 0.6 for a 25-day
period centered around DOY 179. There was a significant negative correlation of growth at both sites with precipitation
centered around DOY 115-125, which overlaps the positive effect of temperature at the beginning of the growing season
(Fig. 5). The remaining significant climate-growth correlations are restricted to very short periods and might be stochastic.





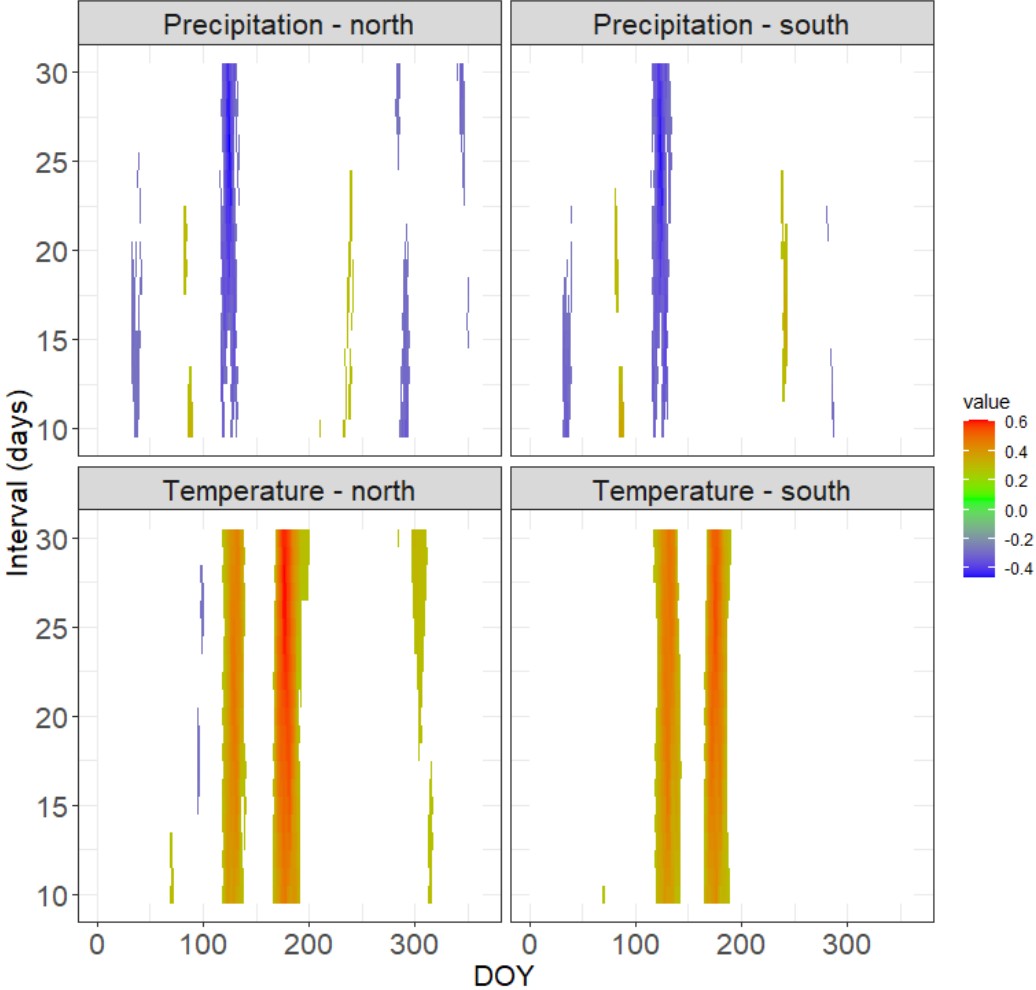

**Figure 5: Climate-growth correlations for tree-ring width chronologies of the north and south-facing site calculated over the period 1961-2013. Only statistically significant correlations are shown, and the time window is centred over the respective days of the year (DOY). The y-axis shows the length of the window of the number of consecutive days for which the correlations were computed. Tree-ring chronologies are shown in Fig. S7.**

## 4 Discussion

We present a study that combines observations of radial stem growth at weekly temporal resolution with the analysis of sub-daily local microclimate and site nutrient availability at cold sites located at the lower part of the treeline ecotone about 150 m below local tree maxima. We found that phenological dates, particularly the onset of wood formation, as well as radial growth in relative terms, were strongly temperature-limited at both sites as expected, but the absolute growth rate was systematically higher on the N-slope than on the S-slope. This is in line with our finding that soils were considerably richer in nutrients on the N-slope, which was also warmer than the S-slope during the nighttime.





### 4.1 Thermal and nutrient limitation of growth in the lower part of the treeline ecotone

We expected to find either the same thermal conditions within tree patches on both slopes or a slightly warmer S-slope
(Paulsen et al., 2001), leading to similar growth onset, duration, and a similar growth rate. However, while we did find that
both sites showed similar thermal conditions and thus similar constraints and timing of beginning and peak growth phases,
the N-slope exhibited a systematically higher seasonal growth rate, albeit significant only in 2014. Consistently, significantly
higher basal area increments were observed for trees growing on the N-slope than on the S-slope.
So far, tree growth at cold sites has been considered predominantly as low temperature-limited (Babst et al., 2019).
Accordingly, we found clear support for the thermal limitation hypothesis as our temperature thresholds for growth
resumption were very similar at both sites (between 3.3°C and 6.4°C depending on the temperature variable) and similar to
the values published elsewhere (Rossi et al., 2007; Körner, 2021). Furthermore, radial stem growth at both sites, expressed as
annual tree-ring width, showed very high sensitivity to variations in temperature during the identical periods of the year:
Ring-width chronologies correlated with temperature mainly during the beginning of the growing season in early May,
which is indicative for the onset of growth in the early growing season (Castagneri et al., 2017; Carrer et al., 2017). The
second period with a significantly positive response of radial growth to temperature is the peak growing season at the end of
June, when the rates of cell division and enlargement culminate (Castagneri et al., 2017; Rossi et al., 2006b). Our data thus
show, in line with current knowledge, that the growing season at cold sites is strongly constrained by temperature, especially
regarding the timing of growth resumption and the timing of the peak rate of cell production during the growing season.
Notably, irrespective of the similar thermal constraints of stem growth on both slopes, the absolute production rate of new
tracheids was systematically higher on the N-slope. Differences between sites were not significant at the 5% probability level
except 2014, probably due to the limited number of sampled trees, though similar or higher in our study than what is a
common standard in wood formation studies (Cuny et al., 2015; Huang et al., 2021). Cell counts were systematically higher
each year on the N-slope, concistent with significantly higher stem growth over the entire lifespan of trees on the N-slope
compared to the S-slope (Fig S5). Factors responsible for the observed differences in growth rate could generally include
differences in microclimate, tree age and size (Rathgeber et al., 2011; Zeng et al., 2017) or accessibility of nutrients. Tree
age was not significantly different between sites in 2012 and 2014 (the years with the greatest differences in cell counts). In
2013, trees were significantly older on the N-slope than on the S-slope, which should be expressed in a lower number of
cells per tree ring on the N-slope due to the age trend in cell number (Lundqvist et al., 2018). However, we observed the
opposite pattern.
It is unclear to what extent microclimatic differences were decisive in our study since we found between-site thermal
differences within or close to the measurement error of the thermistors. The S-slope tended to be slightly more favourable
with respect to air temperature and degree day sums. However, the N-slope was warmer during the night when stem water
potentials are highest with intense cell expansion and division (Zweifel et al., 2021), which may thus benefit stem growth




there. Differences in soil temperature were ambiguous: the S-slope tended to be warmer than the N-slope at the beginning of the growing season, but the N-slope was warmer during the peak growing season, leading to slightly warmer soils on N-slope for the June-September period. Only in the warmest year of the measurement period (2015), the S-slope was substantially warmer.

Overall, our temperature measurements showed that a part of the growing season was characterized by air and soil temperature well above the growth-limiting threshold of >5°C (Körner, 2021), potentially allowing for the influence of other growth-limiting factors. One plausible explanation for the observed growth differences could thus be nutrient availability. Specifically, we assume the better growth performance of trees at the N-slope may be due to a higher availability of base cations and perhaps also P, as can be seen from nutrient concentrations measured both in soils and in needles. Higher concentrations of leaf macronutrients at N-slope could potentially indicate greater sink limit of growth (Hoch and Körner, 2012; Fajardo and Piper, 2017). However, in the light of higher growth rates and concentrations of macronutrients in soils of the N-slope, we interpret this pattern as a consequence of higher uptake of nutrients reflected in the source-driven higher growth rate (Ellison et al., 2019). Our findings underscore the critical role of nutrients on the growth rate and above-ground biomass production (Dobbertin, 2005; Li et al., 2018; Oulehle et al., 2023).

Not surprisingly, in an ecosystem saturated with N (Novotný et al., 2018), the main between-site differences cannot be attributed to N availability but to other nutrients such as Ca, Mg, or P. This applies especially to very acid soils with a pH of around 3, as in our case, where leaching of basal cations is likely (Lucas et al., 2011). In environments rich in C and N, stoichiometric requirements for building new biomass make other nutrients limiting, especially base cations and P (Mellert and Ewald, 2014; Norby et al., 2022; Körner, 2022). So far, most studies accentuated N availability as an important constraint of the growth performance of trees at cold sites (Möhl et al., 2018; Gustafson et al., 2021), but these studies focused on ecosystems not saturated with N. The important role of P for tree growth has also been shown in forest ecosystems near their cold margin (Hagedorn et al., 2020; Ellison et al., 2019). Consistent with the traditional Liebig's law of the minimum (Liebig, 1840), our study highlights the importance of general stoichiometric principles of nutrient requirements for the production of new biomass, which may also play a crucial role in the growth rate of trees at cold sites.

The source of the higher nutrient content in soils of our N-slope is not entirely clear. Metamorphic rocks prevalent on the north-facing slope are generally richer in Mg, Ca and K than granites prevalent on the S-slope, while the content of P is comparable between metamorphites and granites (Czech Geological Survey, 2024). Additionally to weathering, higher nighttime temperatures and slightly higher soil temperatures in the peak growing season on the N-slope may have enhanced decomposition rates and thus indirectly growth performance. Similar relationships between higher soil temperatures, higher soil nutrient content, and enhanced tree growth have been suggested for subarctic treeline ecotones (Sullivan et al., 2015; Dial et al., 2024).



### 4.2 Implications for tree growth in topographically complex cold landscapes

Our data generally support the idea that the thermal differences between high-elevation slopes under forest cover are relatively subtle in the temperate zone (Paulsen and Körner, 2001; Treml and Banaš, 2008, Rita et al., 2021). Additionally, we would like to highlight two striking patterns related to our N-slope and S-slope sites that may be generalizable. First, the duration of direct sunlight during the growing season was longer on the N-slope, probably due to incoming morning and evening sunlight from the northeast and northwest, respectively. This phenomenon might increase with increasing latitudes and be stronger at less steeper slopes. An extreme example is the midnight irradiation of north-facing slopes beyond the polar circle (Kirchhefer, 2000). Second, probably as a consequence, nighttime air temperature was higher on the N-slope compared to the S-slope, leading to equal mean daily air temperature on both sites (the S-slope was warmer during the warmest part of the day). However, since the differences in night temperature were rather high, topographical effects on local air masses ventilation resulting in relatively lower radiative cooling on the N-slope than on the S-slope should also be considered in our case (Barry et al., 1992). Higher nighttime air temperature on the N-slope may also be the cause of warmer soils in some years but with differences close to the measurement error. We conclude that the thermal differences between tree stands growing on supposedly warmer, more growth-favorable south-facing slopes and cooler north-facing slopes are so subtle that they may be overridden by local topography with related relief shading and local circulation.

Faster growth of trees on soils richer in nutrients also implies that the advance of current upper tree limits, which lag behind the pace of warming (Körner and Hiltbrunner 2024), might be faster on fertile soils because seedlings could potentially reach a mature and reproductive age earlier (Dial et al., 2022). This remains to be rigorously tested, although some studies have already suggested greater potential for treeline advancement on fertile soils (Rousi et al., 2018; Gustafson et al., 2021).

### Conclusions

We demonstrated that in the lower part of treeline ecotone of temperate mountains, thermal differences between tree stands growing on supposedly warmer, more growth-favorable south-facing slopes and cooler north-facing slopes are subtle and may be overridden by relief shading and local air circulation. Crucial phases of stem growth, particularly the onset of wood formation and timing of peak growth rate, were constrained by temperature and day length. However, the absolute growth rate was systematically higher on the N-slope with soils considerably richer in nutrients. Our results advocate for a joint effect of nutrient-driven absolute growth rate together with the thermally constrained growth phenology at sites close to the cold range limit of trees. The effect of nutrient availability can be superior to the effect of slope aspect for absolute stem growth rates of *Picea abies* in the lower parts of treeline ecotone. These findings are important for understanding of stem growth trends at treelines which currently often lag behind the pace of warming.



**Data availability**

The data used for the analyses together with R scripts are available here: 10.5281/zenodo.14619874

**Supplement**

This article is accompanied by supplementary material.

**Author contributions**

HK and VT conceptualized the study. HK, and VT performed data processing. TCh and JT contributed to data analyses. HK and VT lead the paper writing. JL, TCh and JT contributed to paper writing (comments and revisions).

**Competing interests**

The authors declare no competing interests.

**Special issue statement**

This article is part of the special issue "Treeline ecotones under global change: linking spatial patterns to ecological processes".

**Acknowledgements**

We appreciate the authority of the Krkonoše National Park for the permission to conduct research and for the logistical support. We further thank Jakub Kašpar and Šárka Zákravská for their help with fieldwork.

**Financial support**

This study was funded by the Czech Science Foundation, grant number 22-26519S. H.K., V.T., J.L. and J.T. were supported by the Johannes Amos Comenius Programme (P JAC), project No. CZ.02.01.01/00/22_008/0004605, Natural and anthropogenic georisks.

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
