# Peer review of "Temperature and nutrient availability influence radial growth of 2 *Picea abies* at opposite slopes in treeline ecotone"

_EGUsphere, 2025_

## Author Response (AR1)

**REVIEWER # 1**

We would like to thank you for the constructive and detailed comments to the manuscript. Below, you can find the point-by-point answers to all your comments.

On behalf of the authors,

Vaclav Treml

1. Since the study only considered single site at each slopes (N-facing and S-facing), the result cannot be generalized, particularly in the mountain landscapes with higher influence of microclimates. Please clarify these issues.

   *Reply: Our replication in terms of number of trees used for xylogenesis and the number of tree-ring series is standard or even higher than in other studies. However, you are true that in terms of general site characteristics of soils and mesoclimate, our replication is low. Throughout the text, we therefore rephrased sentences considering our research settings. I would like to mention that our research was methodologically intensive and hard to apply to multiple sites. We paid attention to place our results in broader context within the Discussion. I hope that we showed that our results match with complementary studies focusing on nutrient limitation of growth rate in cold environments.*

2. There is no any statistical analysis to support the conclusion that soil nutrients drive/modulate their growth.

   *Reply: We did not apply direct statistical model relating tree growth to soil and needle nutrient content. However, we test for differences in soil nutrients between sites and test for differences in tree growth rate. Because the site temperatures are similar and N-site was characterized by significantly higher nutrient content than S-site, we suggest that there is very probable effect of soil nutrients. We are aware of replication limits of our study, therefore we rephrased text towards more careful expressions reflecting some uncertainty.*

3. Precipitation is considered to play a major role in soil formation and development and also tree growth (also mentioned in line 37-39), particularly in the alpine treeline ecotone. However, variation in precipitation between two study sites has not been mentioned. These sites may exhibit difference in precipitation (e.g. windward/leeward effects), which has been largely ignored. At least, soil moisture will be largely different between North-facing and South-facing slope aspects. There are more references on topography-dependent growth responses in the mountain regions. Please clarify.

   *Reply: Thank you for this comment. Precipitation and soil moisture are indeed very important site characteristics. We measured site soil moisture (at microsites beneath tree crowns, L111-112, Fig. S1) and the between-site differences were low and ambiguous (L223-224). We assume that in the mountain region with relatively high precipitation (~1600 mm a year) soils, especially those protected against direct sunlight, retain soil moisture and drought is probably not the important factor of tree growth.*

4. The authors mentioned that soil samples collected during two field campaign (2012-2014 and 2023) were merged to determine soil nutrients, which for me seems to be random.

*Reply: During each sampling campaign we collected 2 to 3 samples which were formed each by several subsamples finally mixed together (L131-132). In this way, each soil sample is similarly representative for entire site. In total, we collected 5 samples which entered into statistical tests. Between-site differences were much greater than within-site differences, that's why we consider our approach as reasonable.*

5. In my understanding, *Picea abies* as a dark forest species, and cannot survive in sunny and south-facing slope. Based on the landscape photo in Fig. 2, it may be not fully source facing slope (east-south or west-south), or the slope is strongly influenced by shade of surroundings, thus have a shade-like environment for spruce forest.

*Reply: It is true that Picea abies is considered as late-successional species. On the other hand, it is a common treeline species in central Europe (marginal parts of the Alps, Carpathians, part of Scandes) so it can profit well also at open sunny sites. At our sites, Picea abies stands are mixed with prostrate Pinus mugo stands, which is also the typical treeline situation in Central Europe.*

Other comments

In my opinion, there is no strong evidence to state that "Radial growth of *Picea abies* is controlled by joint effects of temperature and nutrient availability at the lower part of treeline ecotone". Thus, the title seems to be more ambiguous.

*Reply: Yes, we rephrased the title which now reads as follows: "Temperature and nutrient availability influence radial growth of Picea abies at opposite slopes in treeline ecotone"*

Please clarify the term "lower part of treeline ecotone". It is confusing to me. For the landscape photo, it is not clear whether there is climatic treeline or not in your research sites. It looks to be that trees almost cover the mountain top.

*Reply: I copy below part of the reply to the Rev#2 who raised similar issue. Dark-green tones above the forest stands on the landscape photo are stands of shrubby Pinus mugo. Scattered trees advance roughly 100-150 m above our sites.*

*Reply to Rev#2: In our opinion, our analysis showed that absolute growth is influenced not only by temperatures but also by nutrient availability. Ideally, we would like to relate this finding simply to treeline. However, we are aware that our sites are not located at the very cold margin of trees' fundamental niche (treeline sensu stricto) as our sites are in the lower part of the treeline ecotone and the majority of treelines are currently not in equilibrium with climate. For the sake of exactness, we prefer to state that our findings cannot be generalized for treeline sensu stricto but for the parts of treeline ecotone which are located below the treeline sensu stricto (potential treeline). In fact, this applies to the majority of contemporary treeline ecotones. Based on your comment we inserted sentences justifying our selection of the lower part of treeline ecotone here: Abstract L20,21; Introduction L48-51, L89-91.*

The hypotheses mentioned in abstract and introduction are different.

*Reply: Thank you for this comment. We rephrased text, so that the meaning of the hypotheses in the abstract and introduction is the same.*

In the methodology section, it was mentioned that Kruskal-Wallis test was conducted to investigate the difference in measured soil variables between two sites. Did you find significant difference between sites? Why are their significance not included in the result section and figures? Not only for soil but also for all other variables.

*Reply: Thank you for this comment. Kruskal-Wallis tests were applied to soil data, and to wood phenological data. Differences in growth rates and basal area increments were tested using the confidence intervals. Regarding the temperature and soil moisture, we had to rely only on the comparison considering measuring error of the sensors. For needle nutrient content we worked with a single pooled sample for each site, so any test could be applied. We newly indicate significant differences on box plots using asterisks (Fig. 4, Figs. S2, S3) and insert statements about p<0.05 where tests were applied.*

Which chronology (standard?) was used for the climate-growth relationships? Also, please provide the statistics of the chronology, at least in the supplementary information.

*Reply: The residual chronology was used (described in Methods L188-189). We newly added chronology statistics to the new Table S3.*

Please check for *Picea abies* and make it consistent throughout the text (Italic).

*Reply: Agree, adapted accordingly.*

There are many long sentences with a lot of repetition, which is difficult and tedious to follow. This issue should be carefully revised.

*Reply: We went through the text and tried to shorten long sentences.*

Based on line 97-99, Line 11 "…..contrasting sites located on north and south-facing slopes….." is not true.

*Reply: We leave out the word "contrasting", now L 95.*

Line 254: I suggest not to use the phrase "it is worth mentioning……" here. *Reply: Accepted, deleted*

Line 289: may not be true. *Reply: Yes our assumption turned out to not be true, which is then explained in the following part of Discussion.*

Line 300: also see recent work: Li, et al. 2023. National Science Review 10, nwad182, https://doi.org/10.1093/nsr/nwad182

*Reply: Thank you for this suggestion, we newly refer to this paper.*

Line 302-304, Long sentence with repetition. *Reply: revised accordingly, shortened.*

Line 306: Clarify the term "systematically" *Reply: In the revised version of the manuscript, we reduced usage of this term. By "systematically" we meant that all differences were in the same direction.*

Line 311: Why are there no any references for nutrients? *Reply: Reference provided (L330)*

Line 312-315 It is very difficult to follow. *Reply: Sentences were rephrased (now L330-334)*

Line 367. I suggest to delete "which lags behind the pace of warming". *Reply: accepted, deleted.*

**REVIEWER # 2**

*First of all, thank you for the constructive and detailed comments to the manuscript. Below, you can find the point-by-point answers to all points you raised.*

*On behalf of the authors,*

*Vaclav Treml*

Main comment 1:
The justification for selecting the lower part of the treeline for data collection in your experimental design remains unclear. In various sections of the manuscript, including the title, specific reference is made to "the lower part of the treeline ecotone". The reasons your hypotheses are only applicable to trees within this elevation zone are not specified. The introduction should provide justification for choosing this specific area of the treeline and explain why it is more suitable for conducting the study.

*Reply: In our opinion, our analysis showed that absolute growth is influenced not only by temperatures but also by nutrient availability. Ideally, we would like to relate this finding simply to treeline. However, we are aware that our sites are not located at the very cold margin of trees' fundamental niche (treeline sensu stricto) as our sites are in the lower part of the treeline ecotone and the majority of treelines are currently not in equilibrium with climate. For the sake of exactness, we prefer to state that our findings cannot be generalized for treeline sensu stricto but for the parts of treeline ecotone which are located below the treeline sensu stricto (potential treeline). In fact, this applies to the majority of contemporary treeline ecotones. Based on your comment we inserted sentences justifying our selection of the lower part of treeline ecotone to Abstract (L20,21) and to Introduction (L48-51, L89-91).*

Main comment 2:
The ring-width chronologies have not been sufficiently exploited in the study. Only the results of the correlations between climate and growth have been shown. There is a lack of basic statistics of the chronologies, such as average ring width (or average basal area increment), maximum and average age, serial intercorrelation, mean sensitivity etc. Average basal area increments can be better analysed by comparing the long growth series (100 years?) of the 60 samples taken from north and south sites rather than only the 16 trees (3 cambial years) sampled with Trephor.

*Reply: Thank you for this comment. We newly created Supplementary Table with basic chronology characteristics (Table S3).*

*The results based on mean basal area increments were already in the original version of the manuscript in the Supplementary Materials (former Fig. S4). We newly moved them in the main part of the*

*manuscript (now Figure 5, description at L188-190, 273-274). Basal area increments confirm results*
*derived from microcore sampling.*

Main comment 3:
The estimation of insolation on each site does not consider cloud cover. The discussions lack a section indicating the potential influence of this factor. The results of the correlations between climate and growth show a negative relationship with precipitation in the period DOY 115-125. This relationship is not discussed in the article. The relationship could be due to the negative effect of cloud cover (associated with precipitation), which makes photosynthetic activity less efficient at the beginning of the vegetative period.

*Reply: In our opinion, between-site differences in cloud cover are probably very low with any effect on tree growth. Direct distance between sites across the valley is about 700 m, we cannot assume substantial differences in cloud cover between sites. Negative correlation with precipitation around DOY 115-125 is probably due to anticorrelation with temperature (higher precipitation at that time usually in form of snow delays the beginning of growing season). We added explanatory sentence to Discussion (L312-313).*

Minor comments:

L 9-10: see main comment 1. *Reply: Accepted, rephrased*

L14-16: Please rephrase this sentence. *Reply: Sentence was rephrased, shortened.*

L 78-79 & L 88: Species names in italic. *Reply: We italicized species names throughout the manuscript.*

L 95-96: Is *Picea abies* the only tree species at the treeline? *Reply: Yes, Picea abies dominates treeline ecotone with scarce occurrence of Sorbus aucuparia and Acer pseudoplatanus. Towards higher elevation Picea abies stands are replaced by Pinus mugo shrubland. This information was added to L 96-97.*

Figure 1: Consider moving Figure 1C and 1D in the results section. You explain how you obtain this graph in M&M (ll 121-123). *Reply: Accepted, we split Figure 1 into new Figure 1 (former Fig. 1a,b) and Figure 2 (former Fig. 1c,d).*

L 119-120: Please specify if you consider continuous period or not. *Reply: We consider all days with T>5°C irrespective whether it was continuous or discontinuous period. We inserted a word "all" into the respective sentence (L 121).*

L 136: Please specify if these six trees are the same sampled with Trephor. *Reply: Accepted, rephrased, now L 130-131 ("Sampled trees included those used for xylogenesis research.")*

L 193: "standardization". *Reply: Corrected accordingly*

L 201&210: "°C" instead of "K". *Reply: Changed accordingly including Figures.*

Table 1: Indicate the unit for degree days. *Reply: Here the unit is "degree days" (accumulated °C,K,F above certain threshold), the unit is defined in the name of the variable.*

L294: See main comment 3. It's not clear what is meant by 'entire lifespan of trees' (2012-2014 period or 1961-2013 period?). *Reply: This sentence was rephrased and moved to the next chapter (3.4, now L 271-272). Sentence refers to the new Figure 5 which shows mean basal increments in relation to tree age. ("Mean basal area increments are significantly larger at the N-slope than at the S-slope (Fig. 6) consistently with the higher cell formation rates at the N-slope (Fig. 5).")*